# Mast Cells as a Potential Target of Molecular Hydrogen in Regulating the Local Tissue Microenvironment

**DOI:** 10.3390/ph16060817

**Published:** 2023-05-30

**Authors:** Dmitri Atiakshin, Andrey Kostin, Artem Volodkin, Anna Nazarova, Viktoriya Shishkina, Dmitry Esaulenko, Igor Buchwalow, Markus Tiemann, Mami Noda

**Affiliations:** 1Research and Educational Resource Center for Immunophenotyping, Digital Spatial Profiling and Ultrastructural Analysis Innovative Technologies, Peoples’ Friendship University of Russia Named after Patrice Lumumba, 117198 Moscow, Russia; atyakshin-da@rudn.ru (D.A.); andocrey@mail.ru (A.K.); volodkin-av@rudn.ru (A.V.); anna.nazarova@asticom.net (A.N.); 2Research Institute of Experimental Biology and Medicine, Burdenko Voronezh State Medical University, 394036 Voronezh, Russia; 4128069@gmail.com (V.S.); desaulenko79@gmail.com (D.E.); 3Institute for Hematopathology, Fangdieckstr. 75a, 22547 Hamburg, Germany; mtiemann@hp-hamburg.de; 4Laboratory of Pathophysiology, Graduate School of Pharmaceutical Sciences, Kyushu University, Fukuoka 816-0811, Japan; maminoda39@gmail.com

**Keywords:** molecular hydrogen, reactive oxygen intermediates, mast cells, secretome, specific mast cell proteases, local tissue microenvironment, inflammation

## Abstract

Knowledge of the biological effects of molecular hydrogen (H_2_), hydrogen gas, is constantly advancing, giving a reason for the optimism in several healthcare practitioners regarding the management of multiple diseases, including socially significant ones (malignant neoplasms, diabetes mellitus, viral hepatitis, mental and behavioral disorders). However, mechanisms underlying the biological effects of H_2_ are still being actively debated. In this review, we focus on mast cells as a potential target for H_2_ at the specific tissue microenvironment level. H_2_ regulates the processing of pro-inflammatory components of the mast cell secretome and their entry into the extracellular matrix; this can significantly affect the capacity of the integrated-buffer metabolism and the structure of the immune landscape of the local tissue microenvironment. The analysis performed highlights several potential mechanisms for developing the biological effects of H_2_ and offers great opportunities for translating the obtained findings into clinical practice.

## 1. Introduction

Currently, there is a growing amount of data supporting the participation of reactive oxygen species (ROS) in the initiation and development of inflammation during the disease process at the different age periods of an individual [1,2,3,4,5]. The excessive production of ROS results in oxidative stress, which contributes to the progression of the inflammatory process [6,7,8,9]. In a specific tissue microenvironment, the role of mast cells (MCs) in the regulation of local homeostasis and the integrated-buffer metabolic environment is critical. On the one hand, MCs express a wide range of receptors that provide high-sensitivity mechanisms to form a selective response to external and internal challenges. On the other hand, MCs can selectively secrete various classes of mediators and alternative profiles of cytokines and chemokines, thereby providing targeted effects on the immune and stromal landscapes of a specific tissue microenvironment. MC tools are the three basic classes of mediators—preformed mediators, lipid-derived mediators and multiple cytokines, chemokines, and growth factors formed after MC stimulation for the requisite modification of physiological responses and immune functions [10,11]. MCs are of special significance in the development of a pro-inflammatory background, regulating the state of numerous cells of the immune and stromal landscape, and the extracellular matrix of connective tissue [12,13,14,15,16,17,18,19,20,21]. ROS have the potential to modify and activate MC secretory activity is well known [22,23,24,25,26,27,28,29,30,31,32]. Under oxidative stress, MCs become an essential structural and functional component, and the regulation of this component can affect the integral state of the local tissue microenvironment and its resistance to various challenges. Therefore, it is of interest to search for new molecular agents that can use MC properties to manipulate local inflammation—one of these agents, H_2_, is of special significance. Mechanisms underlying the biological effects of H_2_, including anti-inflammatory, antiapoptotic, neuroprotective, radioprotective, adaptive, homeostatic, etc., have been elucidated, but insufficiently [33,34,35,36,37,38,39,40].

In the present study, MCs are considered potential H_2_ targets in regulating the homeostasis of the integrated-buffer metabolic environment of the extracellular matrix and the immune landscape of a specific tissue microenvironment. Enzymes released by MCs during inflammation, which may be a point of impact for H_2_, are attractive points of discussion. MC activation is accompanied by increased levels of ROS. Considering that MCs or macrophages produce ROS at an early stage of the inflammatory response, H_2_ can be used at the earliest stages of the local inflammatory foci formation. Thus, H_2_ is completely integrated into the list of agents that can influence the activity of MCs indirectly through the effect on ROS.

To date, there are only a few research studies related to the effect of H_2_ on MC biology. H_2_ exposure resulted in a decreased migration and secretory activity of MCs, and a reduced scope of inflammatory reactions in several organs [41,42,43]. In addition, the experiment demonstrated positive effects of H_2_ on the skin dermis remodeling, due to the state of MCs [44]. However, this is clearly insufficient to reveal the actual potential of MCs in developing the biological outcomes of H_2_. This review paper is aimed at focusing the attention of numerous specialists on MCs as a potential target of H_2_ under its therapeutic effects on the pathogenesis of conditions accompanied by acute and chronic inflammation, the formation of which is primarily due to changes in the cell landscape of a specific tissue microenvironment and states of the integrated-buffer metabolic environment of the extracellular matrix.

## 2. Molecular Hydrogen as a Promising Agent for Regulating the State of the Integrated-Buffer Metabolic Environment of the Local Tissue Microenvironment

H_2_ is a diatomic gas consisting of two hydrogen atoms covalently bonded to each other. The cellular bioavailability of H_2_ is extremely high due to its unique physicochemical properties. Its small size, low mass, neutral charge, and nonpolar nature, combined with a high diffusion rate, allow it to freely penetrate cell membranes and diffuse into mitochondria and the nucleus [38,39,45]. The therapeutic and prophylactic effects of H_2_ on various organs have been demonstrated. H_2_ has antioxidant properties, as it immediately counteracts hydroxyl radicals [46] and decreases peroxynitrite levels [37]. Due to its antioxidant action, H_2_ maintains the stability of the genome by a number of markers slowing down the processes of cellular aging, and provides histone modification and telomere maintenance [37]. Apart from this, H_2_ can inhibit inflammatory processes and control the immune system, cell death mechanisms (apoptosis, autophagy, and pyroptosis), the mTOR regulatory pathway, autophagy, apoptosis, and mitochondrial health [37,41,47,48,49,50,51].

Inflammation is induced by the release of pro-inflammatory cytokines, which are produced in the greatest amount by immunocompetent cells, including macrophages, MCs, and neutrophils. The resulting uncontrolled cytokine storm can result in severe conditions with acute or chronic inflammation. Mitochondria generating profuse amounts of the most potential hydroxyl radical •OH appears to be one of the main sources of ROS. H_2_ can selectively neutralize •OH formed in mitochondria and thereby produce its proper effects [52,53,54,55]. As demonstrated, ROS provides NLRP3 inflammasome activation and this, in turn, triggers the pro-inflammatory cytokine production. H_2_ contributes to the inhibition of NLRP3 inflammasome activation by suppressing oxidative stress. This fact is associated with its prophylactic effect related to inflammatory diseases, including the ones in a presymptomatic state [56,57]. Recently, interesting research results have emerged supporting the effective H_2_ application as a new antitumor agent and evidencing its effect on mitigating the side effects of cancer treatment. Notably, mechanisms for the formation of H_2_ effects are not only due to the leveling of •OH, but also the regulation of gene expression [55]. It is reported about the radioprotective effects of H_2_ based on the elimination of •OH due to ionizing radiation [58]. The neuroprotective effects of H_2_ due to antioxidant, anti-inflammatory, anti-apoptotic effects, regulation of autophagy, modulation of mitochondrial function, and the blood–brain barrier have been described as well. As demonstrated, H_2_ has protective effect on ischemic damage to the nervous system, traumatic injuries, subarachnoid hemorrhages, neuropathic pain, neurodegenerative diseases, cognitive dysfunctions, depression, etc. [59]. H_2_ is stated to have diverse options of indirect action, which are mediated by the production of various biologically active substances to form chronic effects [35]. However, the main mechanisms of H_2_ are still far from being revealed, and a further search for tissue and intracellular H_2_ targets is currently required. It should be noted that the effects of molecular hydrogen on a specific tissue microenvironment may depend on the route of intake: drinking hydrogenated water, inhalation with hydrogen, intravenous injections of hydrogen-rich saline, administration of hydrogen water tablets, etc. Despite the excellent ability of H_2_ to diffuse through tissues, it can be assumed that the achievement of its highest concentration at a certain moment is possible in those organs that are a kind of gateway for the entry of H_2_. For example, therapeutic baths with H_2_ will be more beneficial for the treatment of skin diseases, and the use of drinking water enriched with H_2_ is most appropriate for prevention purposes [60]. At the same time, it is quite obvious that if it is necessary to create the highest concentrations of H_2_ in tissues, inhalation with hydrogen might be the most effective way.

## 3. Mast Cells Are Key Regulatory Players in the Organ-Specific Tissue Microenvironment

The state of a specific tissue microenvironment represented by vessels, a cellular component, and an extracellular matrix is critical in the formation of the pathological focus. Each organ has specialized cellular clusters that use proper regulatory mechanisms to maintain local homeostasis. MCs actively participate in the management of cellular cooperations, monitoring most of the key parameters of the cellular microenvironment [10,61,62]. The heterogeneity of MCs determines their specific role for each state of the organ-specific tissue microenvironment. The phenotypic plasticity of MCs creates a great potential for the formation of subpopulations with specialized properties due to epigenetic mechanisms at the level of the local microenvironment. Notably, a balanced regulation of MC adaptation to a specific tissue microenvironment is achieved at each specific temporal point due to various mechanisms, including the effects of tryptase on DNA [63]. On the one hand, the uniqueness of MCs lies in the extraordinary combination of an adapted sensory apparatus to inform meaningful signs of the integrated-buffer metabolic environment, and, on the other hand, of a multifunctional effector apparatus represented by a secretome. The changes existing in the tissue microenvironment are recorded by MCs using multiple receptors, including surface IgG receptors, Toll-like receptors, C-type lectin receptors, retinoic acid-inducible gene-I-like receptors, nucleotide oligomerization domain-like receptors, siglecs, G-protein-coupled receptors, lipid mediator receptors, alarmin receptors, leukocyte Ig-like receptors, cytokine receptors, integrins, tetraspanins, nuclear receptors, etc. [64,65].

In the case of the non-allergic adequate activation of MCs and the challenges of the cellular microenvironment, MCs can secrete, with high selectivity, a variety of biologically active substances, which can be classified as preformed mediators and mediators that resynthesize during the process of MC activation. Pre-accumulated secretome products are represented by biogenic amines (histamine, polyamines, dopamine, serotonin), proteases (chymase, carboxypeptidase A, trypase, cathepsin G, granzyme B, metalloproteinases), and enzymes (kinogenases, heparanase, angiogenin, active caspase-3), including lysosome enzymes (β-hexosaminidase, β-glucuronidase, β-D-galactosidase, arylsulfatase A, cathepsins), proteoglycans (heparin, chondroitin sulfate), cytokines (TNF, IL-4, IL-15, etc.), chemokines (RANTES, eotaxin, IL-8, MCP-1, etc.), growth factors (TGF-β, bFGF, EGF, VEGF, NGF, FGF-2, SCF, PDGF), as well as numerous regulatory peptides (corticoliberin, endorphin, endothelin-1, substance P, vasoactive intestinal peptide, angiogenin, bradykinin, leptin, renin, somatostatin, etc.). The resynthetic products include cell-derived cytokines, growth factors, mitogens, MC-derived chemokines, and various lipid metabolites, in particular prostaglandins and leukotrienes [66,67] (Figure 1).

MC phenotypic plasticity is affected by reactions to specific inflammatory stimuli, this results in their long-lasting polarity effects on adaptive immune responses. Considering the effect of ROS on DNA methylation [68], a pro-inflammatory phenotype of MCs is likely to be formed under certain conditions. Since MCs are long-lived and can be exposed to multiple cycles of activation, they are likely to gain experience in response to repeated stimuli. Such “trained immunity” after exposure to a pathogen has been demonstrated among various immunocompetent cells [69]. The ability of MCs to re-granulate contrasts with the activity of most other immune cells, which undergo apoptosis upon activation [70]. Therefore, MCs have a great potential for tissue remodeling through induced collagen fibrillogenesis, angiogenesis, wound healing, etc., without reaching high quantitative indices per unit area of tissue.

Being an immediate integral part of the developing adaptive and pathological mechanisms through secretome components, MCs serve as an informative marker of disease progression, and represent a promising therapeutic target. Of particular importance are specific MC proteases—tryptase, chymase, and carboxypeptidase A3 [71,72,73]. The secretory mechanisms of proteases and different secretome components provide diverse options for excreting substances having high selectivity into the extracellular matrix, thereby forming a wide range of biological effects [74,75,76,77].

## 4. Mast Cells and Inflammation

MCs play a central topic in the initiation, enhancement, and regulation of inflammation. A significantly increased number of MCs and the activation of degranulation mechanisms in tissues with inflammatory phenomena makes it promising to develop new therapeutic algorithms aimed at targeted regulation of the release of specific mediators. Under inflammation, the number of MCs in the tissue increases dramatically. Activated MCs promptly degranulate, releasing bioactive molecules from secretory granules via numerous mechanisms that ensure the selectivity of their entry into the extracellular matrix [20,76,77]. Activated MCs react during acute and chronic inflammation, in addition to secretome components, and release ROS. Further, the MC phenotype in tissues with inflammation is characterized by an increased density of mass-related G protein-coupled receptor X2 (MRGPRX2). MC activation results in the release of many bioactive compounds, which considers the spatial, temporal, and chemical–physical properties of the local tissue microenvironment, correcting local homeostasis both during the development of inflammation and subsequent restoration of homeostasis to normal values. Thus, each of the points of the intracellular processing of inflammatory mediators in the MCs and post-secretory metabolism are effective targets for the modulation of inflammation by tropic agents. H_2_ can be one such agent.

MC activation is controlled by several receptors: FcɛRI, Toll-like receptors (TLR), KIT receptor, complement receptors C5a and C3a, MRGPRX2, etc. [78,79,80]. Under MC degranulation, tissues release mediators with high pro-inflammatory activity, including specific proteases (tryptase, chymase, and carboxypeptidases); pro-inflammatory cytokines (interleukin (IL)-4,5,6,15), tumor necrosis factor–α (TNF-α); vascular endothelial growth factors (VEGF); biogenic amines (histamine); hydrolases that neutralize pathogens (β-hexosaminidase) [20,71,72,73,81,82]. Notably, MC activation is characterized by an increase in intra- and extracellular ROS [83,84]. Importantly, the prolonged stay of MCs within the inflammation focus leads to chronic activation accompanied by an increased expression of FcɛRI and MRGPRX2 receptors in MCs and, thus, an increased sensitivity to activating signals of a specific tissue microenvironment [67,85]. It goes without saying that this situation will increase the secretion of cytokines, chemokines, and enzymes with pro-inflammatory activity with each circle.

Thus, MCs under normal physiological conditions will differ from MCs in a pathological focus characterized by an inflammatory microenvironment. These data provide promising opportunities for H_2_ targeting specific properties of MCs in tissues with an inflammatory process.

## 5. Reactive Oxygen Species in the Mechanisms of Activation of Mast Cell Secretory Pathways

The release of intracellular and extracellular ROS (ROS_in_ and ROS_ex_, respectively) has an impact on immunocompetent cells, including macrophages, neutrophils, and MCs [31,86,87,88]. It is known that, depending on ROS concentration, they can have a damaging effect on cell macromolecules and, along with this, act as important mediators involved in the regulation of cell growth and differentiation, including various types of cell death [89,90,91,92,93,94]. ROS can directly (as mutagens) and indirectly (as messengers and regulators) affect numerous structural and functional aspects of cell biology. An excess of ROS can result in genomic mutations, but what is more, it can cause irreversible oxidative modification of proteins (oxidation and peroxidation of proteins), lipids and proteoglycans, disrupting their function and contributing to pathological changes as well [41]. Conversely, local ROS at low concentrations are critical as redox signaling molecules in numerous signaling pathways involved in maintaining intracellular homeostasis (MAPK/ERK, PTK/PTP, PI3K-AKT-mTOR, etc.) and in the regulation of key transcription factors (NFκB/IκB, Nrf2/KEAP1, AP-1, p53, HIF-1, etc.) [95]. Therefore, ROS can regulate many cellular functions, including proliferation, differentiation, migration, and apoptosis. The fundamental studies of the molecular mechanisms of ROS biological effects and the potential of their power will be the basis for developing novel therapeutical approaches.

As repeatedly evidenced, MC degranulation attributable to chemical agents (salts of Hg and Au, substance 48/80, Ca^2+^ ionophores, etc.) and physiological stimuli (antigens, neurotrophic growth factor, substance P, etc.) is accompanied by an increased content of ROS in the cytosol [21,23,96,97,98]. Superoxide (O_2_^−^) and hydrogen peroxide (H_2_O_2_) are the two main constituents of ROS in MCs [99,100,101]. After MC activation, ROS_in_ is rapidly released, reaching a peak within a few minutes [21,22,83]. Immune activation upon antigen identification by the IgE-FcεRI complex, as well as non-immunological activation by thapsigargin, ionomycin, and other compounds, is accompanied by an increase in ROS_in_ [83,102,103]. Unlike ROS_in_, there is no consensus regarding the release of ROS_ex_. In addition to ROS_ex_ released exclusively from MCs, mast cells are also exposed to ROS_ex_ action due to the proximity to macrophages and other immunocompetent cells [86,104].

To date, multiple experimental data are a crucial part of ROS in regulating MC degranulation regarding in vitro and in vivo models. ROS_in_ is actively involved in the activation of essential intracellular signaling pathways and can stimulate the production of a number of pro-inflammatory mediators of MC. There are multiple sources of ROS presented in MC mitochondria, which include the electron transport chain, dehydrogenases in the matrix, intermembrane space proteins, monoamine oxidases in the outer membrane, etc. [31,105]. These enzymes form O_2_^−^, which is further a source of H_2_O_2_ and hydroxyl radicals •OH. MCs produce ROS after stimulation of high-affinity IgE receptors (Fc epsilon RI). As known, there are ROS-regulated intracellular and/or plasma membrane Ca^2+^ channels of MCs. In MCs, the activity of the store-operated Ca^2+^ channel (SOC) is regulated by O_2_^−^ and H_2_O_2_. H_2_O_2_ generation is dependent on Src family kinase and phosphatidylinositol-3-kinase activities. Concurrently, O_2_^−^ generation is dependent on Ca^2+^ in the extracellular environment. Thus, the generation of O_2_^−^ and H_2_O_2_ by separating signaling mechanisms reciprocally regulate SOC activity in MCs, which is presented in Ca^2+^ signaling and mediator secretion activity [23,101]. Conditions that promote the formation of ROS in MCs can lead to MC activation by calcium signaling, including hypoxia, allergy, and exposure to aryl hydrocarbon receptor ligands [27,106,107].

ROS produced by NADPH oxidase regulate the pro-inflammatory response of MCs [83]. Regular mitochondrial functions are necessary to provide physiological cellular dynamics, and their dysfunction triggers the development of numerous disorders, including those of the immune system. The impact of external factors on MCs can increase the production of H_2_O_2_ with the participation of mitochondrial complex III, enhancing the secretion of histamine and serotonin by MCs [108]. The use of H_2_ as a substance with antioxidant properties is pathogenetically the most critical mechanism enabling a reduction in MC degranulation activity, and, as a result, theoretically promising potential to decrease the inflammatory background in a specific tissue microenvironment (Figure 2). The proton gradient across the inner mitochondrial membrane is a major driving force for mitochondrial ROS production, and it can be modified by representatives of the mitochondrial uncoupling protein (UCP) family. Of these, UCP2 uncouple oxidative phosphorylation, with concomitant decreases in ROS production, is an effective regulator of MC function [109,110,111]. Recent evidence suggests that UCP2 and mitochondrial translocation regulate MC degranulation [112]. UCP2 not only neutralize ROS, but also prevent their formation, influencing MC degranulation indirectly through an increased concentration of Ca^2+^ [113]. Therefore, UCP2 activators have the potential to reduce the production of mitochondrial ROS [114].

ROS can cause reversible post-translational changes in proteins involved in intracellular signaling. For example, particular proteins contain cysteine residues possessing the ability to be oxidized. Each of these modifications can change the activity of the protein, thereby affecting its function in the signal transduction pathway [41,115].

One of the central events mediated by the influence of ROS and changes in the redox status of the cell is an increased cytoplasmic concentration of Ca^2+^, which are crucial in the mechanisms of MC degranulation [116,117,118]. Concurrently, a change in the intracellular Ca^2+^ concentration, in turn, also affects ROS generation [88]. MC activation comes with ROS production; ROS regulate various signaling pathways, thus providing the release of inflammatory mediators and various cytokine production. Protein tyrosine phosphatases (PTPs) are a superfamily of enzymes that are the main targets for ROS due to an oxidation-susceptible nucleophilic cysteine at their active site [119]. As reported, direct and indirect regulations of class I and II Cys-based protein tyrosine phosphatases (PTEN, LMW-PTP, SHP-2, PTP-PEST, PTP1B, DEP-1, TC45, LAR, etc.) are possible. Notably, SHP-1, SHP-2, and PTEN phosphatases are known to be involved in MC activation [120,121,122]. Phosphatase inhibition by H_2_O_2_ induces phosphorylation of tyrosine residues of β- and γ-subunits of FcεRI, Ca^2+^ influx, and secretory activity of MC [123].

Therefore, redox-regulated protein tyrosine phosphatases may be a target for novel treatment options in allergies or inflammatory diseases using H_2_ [124]. Activation of protein kinase C dependent on ROS can be one of the mechanisms of MC regulation using H_2_ [125]. It should also be noted, that the adapter protein LAT (linker for activation of T cells) can serve as the target for ROS; an interaction with this protein promotes the induction of the FcεRI-dependent pathway of MC activation [126]. Conversely, low-level local ROS are crucial both as redox-signaling molecules in a variety of pathways participating in cellular homeostasis (MAPK/ERK, PTK/PTP, PI3K-AKT-mTOR), and as regulators of key transcription factors (NFκB/IκB, Nrf2/KEAP1, AP-1, p53, HIF-1).

Thus, ROS may have a decisive role in regulating the FcεRI signaling cascade for MC degranulation. As stated, there are many potential MC targets that are sensitive to the effects of ROS. Research investigating the ROS effect on diverse pathways of MC activation, in particular on the FcεRI-dependent pathway, provides great opportunities to further develop and implement into clinical practice drugs tailor-made on the basis of antioxidants and inhibitors resulting from ROS production [16,41]. H_2_ can be used as such an agent, its antioxidant properties are widely discussed [45,52,127]. H_2_ can be used for effective therapeutic action on pathologies associated with MCs, primarily those of an atopic origin. H_2_ applied using various approaches can act as blockers of MC secretory activity, limiting their potential to form a pro-inflammatory ground in a specific tissue microenvironment, and be used to treat multiple systemic inflammatory and allergic disorders [41,42]. Oral ingestion of water with an increased content of H_2_ eliminated an immediate-type allergic reaction in mice [41]. Namely, H_2_ attenuates phosphorylation of the FcεRI-associated Lyn and its downstream signal transduction, thereby inhibiting the NADPH oxidase activity and decreasing H_2_O_2_ generation. The authors also demonstrated that, under an immediate allergic reaction, H_2_ may develop its beneficial effect due to the modulation of different unknown specific signaling pathways [41].

Under the induced intracerebral hemorrhage simulated in male mice, treatment with hydrogen reduced Lyn kinase phosphorylation and tryptase release; decreased MC accumulation and degranulation, which was ultimately accompanied by attenuated blood–brain barrier disruption; reduced cerebral edema; and provided a better neurological status [42]. In addition, the administration of H_2_-rich water was found to reduce skin MC infiltration in the treatment of atopic dermatitis and secretion of pro-inflammatory cytokines such as interleukin IL-1β and IL-33 [43]. Thus, ROS_in_ and ROS_ex_ formed under MC activation in a strict spatio-temporal dependence can be critical for the targeted action of H_2_.

## 6. Preformed Mast Cell Secretome Components—Inflammatory Stimulants

The activation of MCs results in the release of pre-formed mediators, some of which have pronounced pro-inflammatory effects. Moreover, the chronic activation of MCs induces the formation of increased sensitivity to several components of a specific tissue microenvironment due to the adaptation of the receptor apparatus. This provokes a higher secretion of biogenic amines, cytokines, and proteases. The list of inflammatory mediators of MCs is very diverse: specific proteases, histamine, interferon-gamma, IL1-α, IL-1β, -6, -8 and -18, TNFα, LTC4/LTD4, NO, VEGF, NGF, GM-CSF urocortin, LIF, INF-α, IFN-β, etc. [14,128,129,130]. However, in this review, we would like to consider the specific MC proteases in more detail, which are associated with both their high content in MCs and high pro-inflammatory biological activity.

### 6.1. Mast-Cell-Specific Proteases

As reported, there are three specific proteases of human MCs: tryptases, chymase, and carboxypeptidase A3. Studies of the biological role of these enzymes, their activity and inhibition will give rise to novel options in the targeted regulation of inflammations of a specific tissue microenvironment and related disorders.

#### 6.1.1. Tryptase

It is obvious that tryptase release under MC activation, controlled in the time and space of the tissue microenvironment, is the most important event for the development of the inflammatory process. Tryptase is characterized by high biological activity having an impact on the state of multiple cellular and non-cellular components of the tissue microenvironment [73,123,131,132]. Notably, secreted MC proteases promote further intensified degranulation using an autocrine mechanism, and increase the secretory activity of other immunocompetent cells.

The list of conditions with inflammatory processes characterized by MC tryptase involvement is quite long including the cardiovascular, respiratory, digestive, reproductive, nervous system disorders (multiple sclerosis and autoimmune encephalomyelitis among others), as well as the musculoskeletal system, skin disorders, sudden infant death syndrome, sepsis, etc. Tryptase is of particular significance in the mechanisms of oncogenesis, having various target points of application to the molecular pathways of disease progression.

In each condition, tryptase has its proper molecular targets on cells or components of the extracellular matrix. The molecular effects of tryptase at the level of the tissue microenvironment can be differentiated into pro- or anti-inflammatory. Most frequently, tryptase initiates inflammation, resulting in increased permeability of the capillary wall, increasing neutrophils, basophils, eosinophils, and monocytes migration outside the microvasculature. The above tryptase effects are likely to be mediated by the induced kinin, IL-1 and IL-8 formation in the endothelium, which is combined with the modified synthesis of the intercellular adhesion protein ICAM-1. As demonstrated, tryptase is closely involved in the processes of angiogenesis [133,134]. Concurrently, new vessel formation is combined with a marked remodeling of the extracellular matrix, related primarily to the degraded amorphous and fibrous components, secreted growth factors, cytokines and chemokines, and matrix metalloproteinases (MMPs). The joint secretion of MMPs and tryptase is not accidental, since the latter can activate various MMPs within the tissue microenvironment, synthesized by MCs, as well as by other immune and stromal cells in an inactive form [135]. This includes MMP-1, MMP-2, MMP-3, MMP-9, MMP-13, etc. Thus, by activating MMP, tryptase can provide the far-reaching restructuring of the extracellular matrix related to the degraded fibrous component and the ground substance components, including laminin, fibronectin, a number of proteoglycans, etc. Finally, researchers have demonstrated tryptase effects on fibroblastic differon cells; these effects result in the active movement of fibroblastic differon cells, mitotic division, and stimulate the synthesis of collagen proteins. As a result, tryptase effects promote wound healing and provide fibrotic consequences as well.

Tryptase obtains a high affinity for PAR-2 receptors, provoking the progression of the inflammatory processes. PAR-2 receptors localized in different cells of a specific tissue microenvironment can result in mitogenic effects, inducing edema, itching, and other symptoms [135]. In addition, neurogenic inflammation is due to the tryptase-stimulating effect on the PAR-2 receptors of afferent neurons. The persistently increased expression of the PAR-2 receptor in various cells is a crucial regulatory mechanism of tryptase in the potentiation of inflammation. Particularly, this provides functional conditions to strengthen bronchospasm, the secretion of mucus in the mucous membrane in the airways.

An increased expression of PAR-2 receptors causes the progression of arthritis in the cells of specific areas of cartilage [133,136,137]. An increased PAR-2 amount in the soft tissue cells largely aggravates the postoperative course after surgical interventions. In addition, a paramount tryptase effect is an activated secretion of pro-inflammatory mediators by cells of a specific tissue microenvironment, thus creating an increased background content of many cytokines and chemokines.

In the context of the previously described tryptase effects, the progression of allergic reactions is critical. Tryptase results in stimulated histamine secretion from intracellular depots, which, in turn, causes a new increase in the tryptase extracellular level. This contributes to the involvement of new MCs in the process of degranulation, providing conditions for implementing histamine biological effects over a larger area [67,135]. In addition, in the pathogenesis of bronchial asthma, tryptase causes activated proliferation of smooth myocytes, fibroblasts, and cells of the integumentary and glandular epithelium into various membranes of the airway wall. This complex of tryptase bioeffects reduces the patency of the airways, including the one resulting from edematous phenomena in the connective tissue of the bronchial membranes; creates conditions for the hyper-reactivity of the bronchial tree to external stimuli; prolongs the contraction of smooth myocytes; and causes the recruitment of neutrophilic granulocytes and other MCs. Thus, a persistent asthmatic status develops.

Our studies of the MC population properties under various pathological conditions within the specific tissue microenvironment of numerous organs, including cervical cancer, breast, bladder, prostate, etc., demonstrated features of MC localization within a specific tissue microenvironment. Attention is drawn to the contacts of tryptase-positive MCs with pericytes, which can stimulate the formation of an additional pool of requisite cell forms adequately to the current situation within a limited micro-region. Contacts of MCs with leukocytes—in particular, neutrophils, eosinophils, and lymphocytes—are significant [73]. The attachment of MCs to smooth myocytes is a common occurrence. There is morphological evidence of MC contacting nerve endings. The detection of tryptase-positive MCs in the perineurium can not only depend on the state of the extracellular matrix in the nerve trunk, but also affect mechanisms of nerve impulse transmission, which is also related to the formation of pain sensations. MC contacting collagen fibers is frequently detected, although one must consider that the potential of these contacts is simply a way of attachment under MC migration [73].

Recent studies on the essential value of tryptase entry into the nuclei of other cells have demonstrated that protease in the nuclei is able to perform the processing of core histones in the N-terminal tail and thereby participate in the regulation of transcription processes [138,139,140]. DNA molecules have been shown to stabilize the biological activity of tryptase; therefore, protease, even in the absence of heparin and other polyanions of secretory granules, manages to regulate certain events immediately in cell nuclei for sufficiently prolonged periods of time [138,140,141]. This regulatory principle of the epigenetic modification of core histones is a specific function of human MC tryptase [140]. Close colocalization of carboxypeptidase A3 (CPA3) with tryptase, which can be observed by its localization in secretory granules, creates prerequisites for the involvement of CPA3 in several epigenetic effects [71].

#### 6.1.2. Chymase

The biological role of chymase depends on the variants of secretion and is characterized by specific effects on cellular and non-cellular targets of a specific tissue microenvironment. As is known, chymase is closely involved in the atopic and inflammatory processes, angiogenesis and oncogenesis, reconstruction of the extracellular matrix, and restructuring of the organ histoarchitectonics. The number of chymase-positive MCs in the intraorgan population, mechanisms of processing, and secretome degranulation are sensitive criteria for the comprehension of the state of internal organs. Chymase is critical in the signaling molecular–cellular integrative mechanisms of the specific tissue microenvironment, including tumor microenvironment. The analysis of chymase-positive MCs provides great potential to understand physiological and pathological mechanisms occurring in various body systems, including the cardiovascular, respiratory, digestive, musculoskeletal system, the skin, etc. [72,132,142]. Chymase deserves special interest in the light of the fundamental problems of oncology, which supports the search for further study of specific MC proteases in fundamental research and clinical practice. Direct or indirect chymase effects on the smooth muscle tone in the cardiovascular and respiratory organs, vascular microvasculature permeability, immunocompetent cells, fibroblastic differon cells, secretory epithelium, regulation of cell division, growth, differentiation and apoptosis, modulation of the activity of cytokines, chemokines and factors growth, remodeling of the extracellular matrix of a specific tissue microenvironment allow for the expansion of the informative value of histological studies related to a specific internal organ/tissue.

Chymase biological effects developing at the level of a specific tissue microenvironment begin from the moment it enters the extracellular matrix. In humans, the significance of chymase is associated with its active hydrolysis of angiotensin I to angiotensin II and takes part in local and systemic mechanisms for blood pressure maintenance. Angiotensin II, in turn, induces angiogenesis, regeneration, tissue remodeling, and cell growth [67,143]. Chymase can promote the recruitment of immunocompetent cells under various conditions from microvasculature to tissues. There is evidence of an increased accumulation of various inflammatory cells, including eosinophils, neutrophils, lymphocytes, and macrophages, under chymase action [142,144,145,146]. Currently, it is not clear how chymase mediates the recruitment of leukocytes. Chymase acts directly on the activity of diverse components of the extracellular matrix, having a greater potential compared to tryptase [135]. In particular, chymase causes the degradation of fibronectin, laminin, and vitronectin. The mediated biological effects of chymase develop through the activation of collagenase and some MMPs [142,147]. Changes in the structure of fibronectin and vimentin can provoke programmed cell death of the vascular endothelium, or anoikis, due to a change in the state of focal adhesion zones and destruction of the kinase required to maintain the vital activity of epithelial cells [135,147,148]. Researchers investigate variants of degradation of tight junction proteins, leading to increased permeability of the endothelium, intestinal epithelium, and epidermis [144,149,150].

The pro-inflammatory effects of chymase are linked to the activation of cytokines and growth factors, such as IL-1β, IL-8, IL-18, TRF-β, endothelin-1 and -2, neutrophil-activating protein-2, etc., which results in recruitment of granulocytes, lymphocytes, and monocytes into the tissue microenvironment [142,151,152]. The correlation of chymase activity with the progression of diseases associated with inflammation has been demonstrated, namely, respiratory system disorders and kidneys pathologies, systemic scleroderma, arthritis, psoriasis, chronic trophic leg ulcers, diabetic nephro- and retinopathy, metabolic diseases, aortic aneurysm, experimental autoimmune encephalomyelitis and multiple sclerosis, etc. [15,16,142,146,147,151,153,154,155,156]. Chymase can cause the degradation of contacts and structures that ensure the strength of the attachment of epitheliocytes to each other and the basement membrane, resulting in a decreased barrier function of the epithelial layer [157]. By potentiating histamine effects relating to an increased number of vesicles and blisters on the skin, chymase promotes the breakdown of the components of the airway epithelial glycocalyx, stimulates the secretory activity of the glands of the bronchial mucosa and induces the synthesis of IgE. Chymase is an inducer of further MC degranulation and promotes the entry of histamine into the extracellular matrix. Concurrently, chymase effects such as degraded TNF-α, bradykinin, complement component C3a, eotaxin, preproendothelin-1, substance P, vasointestinal peptide, kallikrein, calcitonin-gene-related peptide and a number of ILs: IL-1β, -5, -6, -13, -18, -33 are also known [135,148,151,158].

The role of chymase in collagen biogenesis is critical. On the one hand, chymase induces increased mitotic activity of fibroblasts along with their biosynthetic potential. In addition, chymase is involved in the procollagen molecule modification, providing the formation of collagen fibrils [135,159]. The conducted research [160] demonstrates that MCs actively participate in fibrillogenesis; this is expressed by an inductive effect on the fibrous component formation of the tissue microenvironment in the pericellular space of fibroblastic differon cells. The reticular fibers or points of fibrillogenesis initiation were also near the MC plasmalemma [160].

#### 6.1.3. Carboxypeptidase A3

Despite the abundance of CPA3 in MCs, there is currently a lack of knowledge about its biological effects compared to other specific MC proteases—tryptase and chymase [71,132,161]. As reported, the best-known fact is the CPA3 involvement in innate immunity; namely, the involvement of protease in the body’s defense against snake venoms and certain toxins has been demonstrated [162,163,164]. The functional value of the CPA3 complex with chymase has been repeatedly studied, including aspects of its more efficient substrate degradation under protease colocalization [142,165]. In addition, chymase and CPA3 are able to interact in the enzymatic degradation of angiotensin II [166], where each of the enzymes has proper catalytic activity towards specific substrate proteins.

The sufficiently proximate cytological colocalization of chymase and CPA3 can persist after MC secretion and participate in the conveyor cleavage of various peptide targets of the extracellular matrix, requiring the synchronous presence of endo- and exopeptidases. One can assume that CPA3 is critical in the processing and remodeling of the extracellular substance fibrous component. Conversely, the CPA3-chymase complex can cause increased proliferative activity of fibroblasts, as well as their biosynthetic possibilities. However, this entire complex or proteases separately are able to participate in the transformation of procollagen molecules, inducing collagen fibril formation [135,159]. MCs have an active role in the mechanisms of fibrillogenesis, which is expressed by an inductive effect on the formation of the collagen fibers of the tissue microenvironment [160]. It is demonstrated that reticular fibers or points of fibrillogenesis initiation are near the plasmalemma of MCs [160]. These aspects are indirectly supported by studies simulating adhesive processes in the abdominal cavity in laboratory animals, and in the studies of lungs and kidneys exposed to chronic inflammation or fibrosis [167,168].

The incomplete restoration of bone tissues was detected in Cpa3Cre+/line mice with Cpa deficiency in MCs due to a decreased number of the intraorganic MC population, delayed revascularization, accumulation and mineralization of the intercellular substance of the newly formed osseoid, and modified activity of osteoclasts and macrophages [169]. Changes detected in CPA3 expression in multiple disorders highlight numerous points of protease application; their further studies manage to identify new molecular potentials for targeted therapy to increase therapeutic effectiveness [170,171,172,173].

As stated, tryptase has epigenetic effects due to its action on the state of nuclear histones and DNA stabilization [63,140,141]. Close colocalization of CPA3 with tryptase provides special interest in the extent of exopeptidase involvement in these effects.

## 7. Concluding Remarks

Thus, MCs are closely involved in the development of the biological effects of H_2_ at the level of a specific tissue microenvironment and may be involved in the development of its anti-allergic, anti-inflammatory, anti-apoptotic, immunomodulatory, vasotropic, and extracellular matrix remodeling effects. The existing H_2_ therapy may have targeted properties relating to the functional potential of MC mediators and the gradual modulation of the receptor apparatus, due to which, a different phenotype of the MC population is formed. Since MCs are critical in the pathogenesis of several inflammatory diseases, they represent a promising target in the development of therapeutic approaches to modify the molecular portrait of the specific tissue microenvironment, including the extracellular matrix and the immune landscape; this can be applied in the management of various conditions associated with the development of acute and chronic inflammation. Further studies of the biological effects of H_2_ mediated by MCs will create new pathways in personalized and preventive medicine.

## Figures and Tables

**Figure 1 pharmaceuticals-16-00817-f001:**
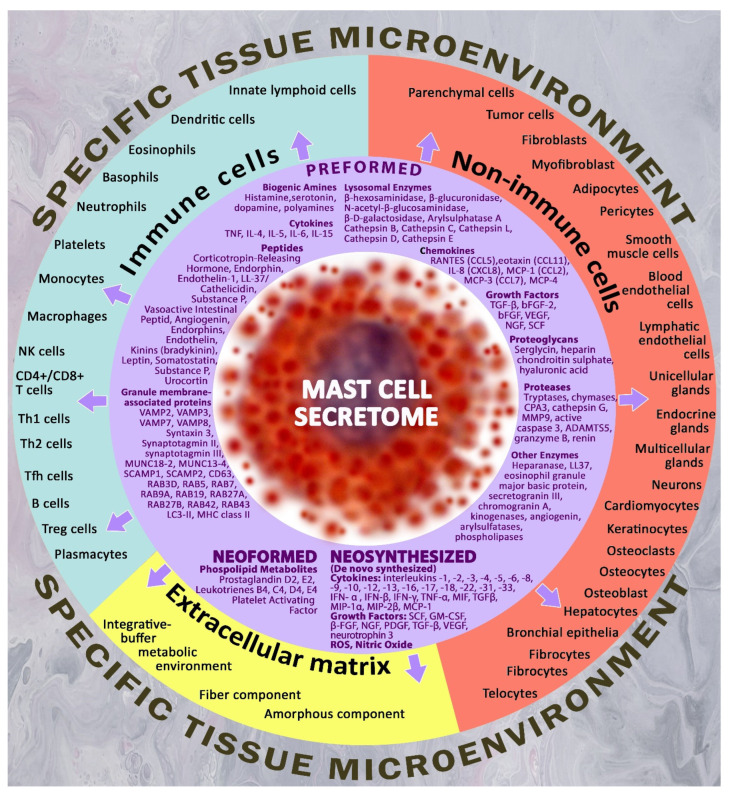
Targets for Mast Cell Secretome Components in a Specific Tissue Microenvironment. The figure shows the main targets of the specific tissue microenvironment for mast cells, including non-immune cells, immunocompetent cells and extracellular matrix targets. The arsenal of biologically active substances of MC for secretion is represented by a wide range of preformed, neoformed and neosynthesized mediators, which allows for the highly selective regulation of the state of cellular and extracellular targets within the determined physiological constants. Thus, by influencing the biosynthetic and secretory activities of mast cells, molecular hydrogen is able to indirectly regulate the key parameters of the local microenvironment within limited tissue compartments and niches.

**Figure 2 pharmaceuticals-16-00817-f002:**
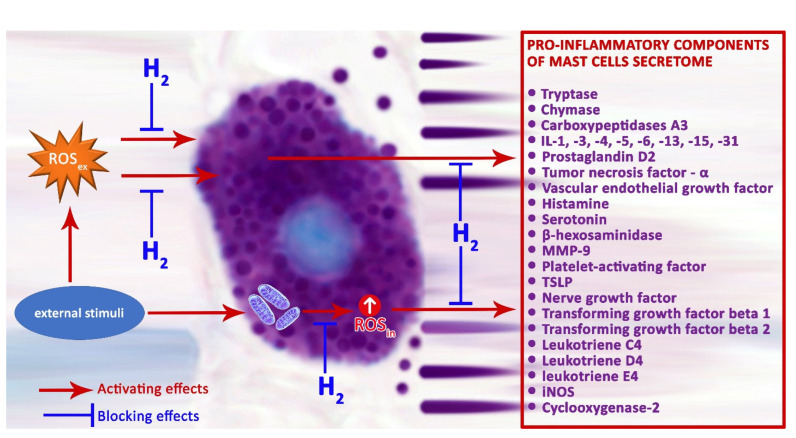
Possible mechanisms of H_2_ regulation on the secretory pathways of pro-inflammatory mast cell mediators.

## Data Availability

All data and materials are available upon reasonable request. Address to I.B. (email: buchwalow@pathologie-hh.de) or M.T. (email: mtiemann@hp-hamburg.de) Institute for Hematopathology, Hamburg, Germany.

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
