# Peer review of "Mast Cells as a Potential Target of Molecular Hydrogen in Regulating the Local Tissue Microenvironment"

_pharmaceuticals, 2023, doi:10.3390/ph16060817_

Round 1
Reviewer 1 Report
The presented article by authors Atiakshin et al. summarizes the current scientific knowledge about the mechanisms of molecular hydrogen action in biological systems with a focus on the mast cells as a potential target for molecular hydrogen. Till now, the exact mechanism of molecular hydrogen has not been fully elucidated. Since it is a new potential therapeutic substance with a pluripotent effect in many diseases, this topic is actual and attractive.
This is a very well-written, comprehensive, easy-to-read, up-to-date, and interesting review. Even though there are only a few research studies related to the effect of molecular hydrogen on mast cell biology, the authors formulated suitable hypotheses that can form the basis for further research in this area.
In conclusion, I recommend this article to be accepted for publication in the present form in this journal.
Author Response
Reviewer #1
Rev #1 Comments and Suggestions for Authors
The presented article by authors Atiakshin et al. summarizes the current scientific knowledge about the mechanisms of molecular hydrogen action in biological systems with a focus on the mast cells as a potential target for molecular hydrogen. Till now, the exact mechanism of molecular hydrogen has not been fully elucidated. Since it is a new potential therapeutic substance with a pluripotent effect in many diseases, this topic is actual and attractive.
This is a very well-written, comprehensive, easy-to-read, up-to-date, and interesting review. Even though there are only a few research studies related to the effect of molecular hydrogen on mast cell biology, the authors formulated suitable hypotheses that can form the basis for further research in this area.
In conclusion, I recommend this article to be accepted for publication in the present form in this journal.
Authors:
The authors thank the referee for a careful analysis of our article.
Reviewer 2 Report
Comments and suggestions
1. In the abstract authors wrote; Knowledge of the biological effects of molecular hydrogen (H2), hydrogen gas, is constantly expanding giving a reason for optimism for several healthcare practitioners regarding management of multiple diseases, including socially significant ones. What is the meaning socially significant ones? need clarification.
2. The aim of this study focus not clear? I think abstract should be rewrite.
3. In the introduction; This review paper is aimed at focusing attention of numerous specialists on MCs as a potential target of H2 under its therapeutic effects on the pathogenesis of conditions accompanied by acute and chronic inflammation. But in the abstract authors mentioned In this review, we focus on mast cells as a potential target for H2 at the level of a specific tissue microenvironment. It should be consistent.
4. For abbreviation first time use abbreviation with short form then use short form throughout the manuscript.
5. Figure 1 is not clear for visualization.
6. Section 2 should be expanded with relevant studies if your study focuses on it.
7. Section 4 Mast cells and inflammation. Authors can show the mechanism by drawing figure.
8. Please check section 5 some information is overlapping with the previous action.
9. Conclusion should be more specific not details.
10. Please check all syntax, typo, and grammatical errors throughout of the manuscript.
It needs to improve English of this manuscript. Although, it has written well in general.
Author Response
Reviewer #2
The authors thank the referee for a careful analysis of our article and important remarks, to which we tried to give exhaustive answers.
2.1. In the abstract authors wrote; Knowledge of the biological effects of molecular hydrogen (H2), hydrogen gas, is constantly expanding giving a reason for optimism for several healthcare practitioners regarding management of multiple diseases, including socially significant ones. What is the meaning socially significant ones? need clarification.
Authors:
The authors thank the referee for the remark.Indeed, we missed this moment.For correction, we have made an addition to this text: malignant neoplasms, diabetes mellitus, viral hepatitis, mental and behavioral disorders.
Previously: Knowledge of the biological effects of molecular hydrogen (H2), hydrogen gas, is constantly expanding giving a reason for optimism for several healthcare practitioners regarding management of multiple diseases, including socially significant ones.
Corrected: Knowledge of the biological effects of molecular hydrogen (H2), hydrogen gas, is constantly expanding giving a reason for optimism for several healthcare practitioners regarding management of multiple diseases, including socially significant ones (malignant neoplasms, diabetes mellitus, viral hepatitis, mental and behavioral disorders).
2.2. The aim of this study focus not clear? I think abstract should be rewrite.
Authors: “The abstract stated the following: “In this review, we focus on mast cells as a potential target for H2 at the level of a specific tissue microenvironment.”
However, due to the recommendations of the reviewer, we have made here corrections: " In this review, we focus on mast cells as a potential target for H2 at the specific tissue microenvironment level .
2.3. In the introduction; This review paper is aimed at focusing attention of numerous specialists on MCs as a potential target of H2 under its therapeutic effects on the pathogenesis of conditions accompanied by acute and chronic inflammation. But in the abstract authors mentioned In this review, we focus on mast cells as a potential target for H2 at the level of a specific tissue microenvironment. It should be consistent.
Answer: The specific tissue microenvironment in any organ is the scene of all pathogenetic changes, which, after reaching a certain extent, can already manifest themselves as obvious clinical signs or symptoms. In particular, detectable inflammatory syndromes in many diseases of various organs are formed primarily at the level of the immune and stromal landscapes of the local tissue microenvironment. Therefore, in the abstract, we would not like to limit the prospect of mast cells as a target for molecular hydrogen only in the development of inflammation. In accordance with the reviewer's point of view, we have changed the wording in the introduction:
Originally: "This review paper is aimed at focusing attention of numerous specialists on MCs as a potential target of H2 under its therapeutic effects on the pathogenesis of conditions accompanied by acute and chronic inflammation."
Corrected: “This review paper is aimed at focusing attention of numerous specialists on MCs as a potential target of H2 under its therapeutic effects on the pathogenesis of conditions accompanied by acute and chronic inflammation, the formation of which is primarily due to changes in the cell landscape of a specific tissue microenvironment and states of the integrative-buffer metabolic environment of the extracellular matrix”.
2.4. For abbreviation first time use abbreviation with short form then use short form throughout the manuscript.
Authors: Corrected.
2.5. Figure 1 is not clear for visualization.
Authors: In accordance with the reviewer's remark, we made a caption for figure No. 1: “The figure shows the main targets of the specific tissue microenvironment for mast cells, including non-immune cells, immunocompetent cells and extracellular matrix targets. The arsenal of biologically active substances of MC for secretion is represented by a wide range of preformed, neoformed and neosynthesized mediators, which allows highly selective regulation of the state of cellular and extracellular targets within the determined physiological constants. Thus, by influencing the biosynthetic and secretory activities of mast cells, molecular hydrogen is able to indirectly regulate the key parameters of the local microenvironment within limited tissue compartments and niches.
2.6. Section 2 should be expanded with relevant studies if your study focuses on it.
Authors: The authors thank the referee for an important recommendation.But in our study, we do not pretend to provide a comprehensive analysis of this issue in this review.The information in Section 2 is presented in order to draw the reader's attention to the fundamental possibility of the influence of molecular hydrogen on a number of constants of the integrated-buffer metabolic environment of the local tissue microenvironment.This helps to achieve the main goal of the review - to reveal and suggest the possible regulatory properties of molecular hydrogen in relation to mast cells, and, accordingly, the development of mediated effects.
2.7. Section 4 Mast cells and inflammation. Authors can show the mechanism by drawing figure.
Authors: Thanks to the reviewer for the comment. Many of the biological effects of mast cells on the development of inflammation on a specific tissue microenvironment are reflected in Figure 2. We hope this will be enough.
2.8. Please check section 5 some information is overlapping with the previous action.
Authors: Indeed, it is. This is due to the fact that Reactive oxygen species play an important role in the development of inflammation. In this section, we take a closer look at this issue. In addition, for ease of understanding, we have prepared Figure 2.
2.9. Conclusion should be more specific not details.
Authors: We've removed a few sentences from the "Concluding Remarks" section to make the information more specific:
Orginally: Thus, MCs are closely involved in the development of biological effects of H2 at the level of a specific tissue microenvironment and may be involved in the development of its anti-allergic, anti-inflammatory, anti-apoptotic, immunomodulatory, vasotropic, and extracellular matrix remodeling effects. The existing H2 therapy, in fact, may have targeted properties relating to the functional potential of MC mediators and gradual modulation of the receptor apparatus, due to which a different phenotype of the MC population is formed. Since MCs are critical in the pathogenesis of several inflammatory diseases, they represent a promising target in the development of therapeutic approaches. It should be considered that modulation of the MC phenotype under chronic inflammation causes formation of "faulty" circles of pathogenesis, progression of tissue damage and an increased severity of the disease, significantly reducing the quality of life. Activated MCs produce ROS, which can be used to control the immune response at the earliest stages of the disease. Thus, H2 effect on MCs allows targeting the disease at the very beginning and with pronounced progression due to the regulatory effects of mediators. Mobile regulation of the ROS level in MCs through the systemic use of H2 supports a wide range of potentials to modify the phenotype of the integrated-buffer metabolic environment of the extracellular matrix and the immune landscape of the specific tissue microenvironment; this can be applied in the management of various conditions associated with the development of acute and chronic inflammation. Further studies of the biological effects of H2 mediated by MCs will open new pages in personalized and preventive medicine.
After correction: Thus, MCs are closely involved in the development of the biological effects of H2 at the level of a specific tissue microenvironment and may be involved in the development of its anti-allergic, anti-inflammatory, anti-apoptotic, immunomodulatory, vasotropic, and extracellular matrix remodeling effects. The existing H2 therapy may have targeted properties relating to the functional potential of MC mediators and the gradual modulation of the receptor apparatus, due to which a different phenotype of the MC population is formed. Since MCs are critical in the pathogenesis of several inflammatory diseases, they represent a promising target in the development of therapeutic approaches to modify the molecular portrait of the specific tissue microenvironment, including the extracellular matrix and the immune landscape; this can be applied in the management of various conditions associated with the development of acute and chronic inflammation. Further studies of the biological effects of H2 mediated by MCs will create new pathways in personalized and preventive medicine.
- Please check all syntax, typo, and grammatical errors throughout of the manuscript.
Authors: Correspondingly corrected.
- Comments on the Quality of English Language
It needs to improve English of this manuscript. Although, it has written well in general.
Authors: Thank you for the comments. We used MDPI's Author Services for English Editing [English ID: english-66171].
Reviewer 3 Report
Review of the paper entitled “Mast Cells as a Potential Target of Molecular Hydrogen in Regulating the Local Tissue Microenvironment” by Dmitri Atiakshin, Andrey Kostin, Artem Volodkin, Anna Nazarova, Viktoriya Shishkina, Dmitry Esaulenko, Igor Buchwalow, Markus Tiemann and Mami Noda.
The first reports point to the therapeutic potential of molecular hydrogen come from the first decade of the 21st century. Since then, molecular hydrogen therapy has been widely concerned and researched. In this review, the authors focus on mast cells as a potential target for molecular hydrogen at the level of a specific tissue microenvironment.
It is an interesting and well written article.
The authors could extend their article with a short information on the method of administration and the dose of molecular hydrogen. It is generally known that there are several ways to deliver molecular hydrogen to the organism. These are: drinking hydrogenated water, inhalation with hydrogen, intravenous injections of hydrogen-rich saline, administration of hydrogen water tablets. Which of these methods, according to the authors, allows you to achieve the best therapeutic effect. In what doses of hydrogen is and should be administered.

Author Response
Reviewer #3
Comments and Suggestions for Authors
Review of the paper entitled “Mast Cells as a Potential Target of Molecular Hydrogen in Regulating the Local Tissue Microenvironment” by Dmitri Atiakshin, Andrey Kostin, Artem Volodkin, Anna Nazarova, Viktoriya Shishkina, Dmitry Esaulenko, Igor Buchwalow, Markus Tiemann and Mami Noda.
The first reports point to the therapeutic potential of molecular hydrogen come from the first decade of the 21st century. Since then, molecular hydrogen therapy has been widely concerned and researched. In this review, the authors focus on mast cells as a potential target for molecular hydrogen at the level of a specific tissue microenvironment.
It is an interesting and well written article.
3.1. The authors could extend their article with a short information on the method of administration and the dose of molecular hydrogen. It is generally known that there are several ways to deliver molecular hydrogen to the organism. These are: drinking hydrogenated water, inhalation with hydrogen, intravenous injections of hydrogen-rich saline, administration of hydrogen water tablets. Which of these methods, according to the authors, allows you to achieve the best therapeutic effect. .
The authors thank the referee for the analysis of our article. According to the wishes of the author, a corresponding addition was made to section “2. Molecular hydrogen as a promising agent for regulating the state of the integrated-buffer metabolic environment of the local tissue microenvironment”:
“It should be noted that the effects of molecular hydrogen on a specific tissue microenvironment may depend on the route of intake: drinking hydrogenated water, inhalation with hydrogen, intravenous injections of hydrogen-rich saline, administration of hydrogen water tablets, etc. Despite the excellent ability of H2 to diffuse through tissues, it can be assumed that the achievement of its highest concentration at a certain moment is possible in those organs that are a kind of gateway for the entry of H2. For example, therapeutic baths with H2 will be more beneficial for the treatment of skin diseases, and the use of drinking water enriched with H2 is most appropriate for prevention purposes (Ichihara G 2021) [60]. At the same time, it is quite obvious that if it is necessary to create the highest concentrations of H2 in tissues, inhalation with hydrogen might be the most effective way.”
[60] Ichihara, G.; Katsumata, Y.; Moriyama, H.; Kitakata, H.; Hirai, A.; Momoi, M.; Ko, S.; Shinya, Y.; Kinouchi, K.; Kobayashi, E.; et al. Pharmacokinetics of hydrogen after ingesting a hydrogen-rich solution: A study in pigs. Heliyon 2021, 7, e08359, doi:10.1016/j.heliyon.2021.e08359.
Round 2
Reviewer 2 Report
Good luck for your manuscript.
Minor editing of English language required